# Stack More Layers Differently:
# High-Rank Training Through Low-Rank Updates

## Abstract

Despite the dominance and effectiveness of scaling, resulting in large networks with hundreds of billions of parameters, the necessity to train overparametrized models remains poorly understood, and alternative approaches do not necessarily make it cheaper to train high-performance models. In this paper, we explore low-rank training techniques as an alternative approach to training large neural networks. We introduce a novel method called ReLoRA, which utilizes low-rank updates to train high-rank networks. We apply ReLoRA to pre-training transformer language models with up to 350M parameters, and demonstrate comparable performance to regular neural network training. Furthermore, we observe that the efficiency of ReLoRA increases with model size, making it a promising approach for training multi-billion-parameter networks efficiently. Our findings shed light on the potential of low-rank training techniques and their implications for scaling laws.[1]

## 1   Introduction

Over the past decade, the machine learning field has been dominated by the trend of training increasingly overparametrized networks or adopting the "stack more layers" approach [32, 21, 27]. The definition of a large network has evolved from models with 100 million [46, 39] to hundreds of billions [8, 12] of parameters, which has made computational costs associated with training of such networks prohibitive to most of the research groups. Despite this, the necessity to train models which can have orders of magnitude more parameters than the training examples [8, 12, 16], is poorly understood theoretically [25, 4, 60].

Alternative approaches to scaling, such as more compute-efficient scaling optima [22], retrieval-augmented models [28, 7], and the simple approach of training smaller models for longer [50], have offered new interesting trade-offs. However, they do not bring us closer to understanding why we need overparametrized models and rarely democratize the training of these models. For example, training RETRO [7] requires a complex training setup and infrastructure capable of quickly searching over trillions of tokens, while training LLaMA-6B [50] still requires hundreds of GPUs.

In contrast, approaches like zero-redundancy optimizers [43], 16-bit training [37], 8-bit inference [14], and parameter-efficient fine-tuning (PEFT) [33] have played a crucial role in making large models more accessible. Specifically, PEFT methods have enabled fine-tuning of billion-scale language or diffusion models on consumer hardware. This raises the question: Can these approaches also benefit pre-training?

On one hand, pre-training is exactly the step that allows for small modifications to the network to adapt it to new tasks. Aghajanyan et al. [1] demonstrated that the rank of the changes required

---

[1]The code is provided with the supplementary material of the submission.

Submitted to 37th Conference on Neural Information Processing Systems (NeurIPS 2023). Do not distribute.

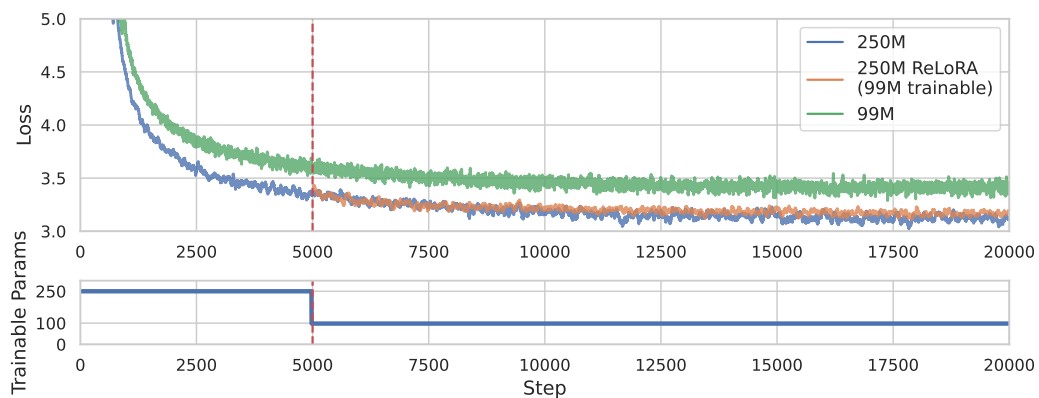

Figure 1: ReLoRA learns a high-rank network through a sequence of low-rank updates. It outperforms networks with the same trainable parameter count and achieves similar performance to training a full network at 100M+ scale. The efficiency of ReLoRA increases with the model size, making it a viable candidate for multi-billion-parameter training.

to learn a task decreases the more you pre-train the network. On the other hand, multiple studies have demonstrated the simplicity of features extracted and utilized by language and vision models, along with their low intrinsic dimensionality [31, 17, 42, 47]. For instance, attention patterns in transformers [51] often exhibit a small rank, which has been successfully leveraged to develop more efficient variants of attention [52, 11]. Moreover, overparametrization is also not necessary for training. The Lottery Ticket Hypothesis [17] empirically demonstrates that during initialization (or early in training [18]), there exist sub-networks – winning tickets – that when trained in isolation reach the performance of the full network.

In this study, we focus on low-rank training techniques and introduce ReLoRA that uses low-rank updates to train a high-rank network. We empirically demonstrate that ReLoRA performs a high-rank update and achieves performance similar to regular neural network training. The components of ReLoRA include initial full-rank training of the neural network (similar to Frankle et al. [18]), LoRA training, restarts, a jagged learning rate schedule, and partial optimizer resets. We evaluate ReLoRA on transformer language models up to 350M parameters. We chose to focus on autoregressive language modeling, as this approach has demonstrated its universality in most of the applications of neural networks [41, 56, 3, 35, 10]. Finally, we observe that the efficiency of ReLoRA increases with model size, making it a viable option for efficient training of multi-billion-parameter networks.

*Each experiment in this study has used no more than 8 GPU days of compute.*

## 2   Related work

**Scaling versus Efficiency**   The relationship between overparametrization and neural network trainability and generalization has been extensively studied [59, 5, 17, 38, 47], yet it remains a mystery [60]. Moreover, scaling laws [27, 19, 22, 30, 2] demonstrate a simple and strong power-law dependence between network size and its performance across a variety of modalities. This finding not only supports overparametrization but also encourages the training of extraordinarily resource-intensive neural networks [8, 12, 16]. Nonetheless, the Lottery Ticket Hypothesis [17, 18] suggests that overparametrization could, in principle, be minimized. Specifically, it shows that *early in training, subnetworks exist that can be trained to achieve the performance of the full network (winning tickets).*

**Parameter-efficient fine-tuning**   Aghajanyan et al. [1] found that pre-training reduces the amount of change to the network, or its intrinsic dimensionality, to learn a new task through fine-tuning. I.e., larger networks or networks pre-trained on more data require smaller modifications in terms of the rank of the range to learn a new task. This explains the success of parameter-efficient fine-tuning methods [33] and has also motivated the development of low-rank fine-tuning methods such as LoRA [23] and Compacter [36].

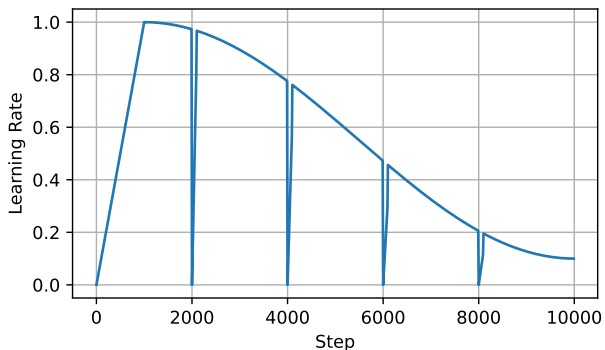

Figure 2: Jagged cosine scheduler used in ReLoRA. On every ReLoRA reset, we set the learning rate to zero and perform a quick (50-100 steps) learning rate warmup back to the cosine schedule.

**Low-rank neural network training**   Training low-rank representations has been explored in the context of CNN compression, regularization, and efficient training [24, 26, 49, 44, 34, 57]. However, most of these methods are either specific to CNNs, do not scale well, or have not been evaluated on large transformers [51] with hundreds of millions of parameters, which can benefit greatly from efficient training. While transformers have been shown to have a low-rank internal dimensionality and representations [1, 52], the study by Bhojanapalli et al. [6] demonstrated that the low rank of key and query projections in multi-head attention bottlenecks the performance of transformers. Our own experiments (Section 3) also demonstrate that low-rank transformers perform significantly worse compared to the full-rank baseline and ReLoRA.

## 3   Method

Let's start by revisiting linear algebra-101. In particular, we are interested in the rank of the sum of two matrices:

$$\text{rank}(A + B) \leq \text{rank}(A) + \text{rank}(B). \tag{1}$$

This bound on the rank of the sum is tight: for a matrix $\mathbf{A}, \text{rank}(\mathbf{A}) < dim(\mathbf{A})$, there exists $\mathbf{B}$, $\text{rank}(\mathbf{B}) < dim(\mathbf{B})$ such that sum of the matrices has a higher rank than either $\mathbf{A}$ or $\mathbf{B}$. We want to exploit this property to make a flexible parameter-efficient training method. We start with LoRA [23] which is a parameter-efficient fine-tuning method based on the idea of low-rank updates. LoRA can be applied to any linear operation parametrized through $W \in \mathbb{R}^{m \times n}$. Specifically, LoRA decomposes the weight update $\delta W$ into a low-rank product $W_A W_B$ as shown in Equation 2, where $s \in \mathbb{R}$ is a fixed scaling factor usually equal to $\frac{1}{r}$.

$$\begin{aligned} \delta W &= s W_A W_B \\ W_A &\in \mathbb{R}^{\text{in} \times r}, W_B \in \mathbb{R}^{r \times \text{out}} \end{aligned} \tag{2}$$

In practice, LoRA is usually implemented by adding new trainable parameters $W_A$ and $W_B$, which could be merged back into the original parameters after training. Thus, even though Equation 1 allows the total update over training time $\sum_t \delta W_t$ to have a higher rank than any of the individual matrices, LoRA implementations are restricted by the rank $r = \max_{W_A, W_B} \text{rank}(W_A W_B)$.

If we could restart LoRA, meaning we merge $W_A$ and $W_B$ during training and reset the values of these matrices, we could increase the total rank of the update. Doing this multiple times brings the total neural network update to

$$\Delta W = \sum_{t=0}^{T_1} \delta W_t + \sum_{t=T_1}^{T_2} \delta W_t + \cdots + \sum_{t=T_{N-1}}^{T_N} \delta W_t = s W_A^1 W_B^1 + s W_A^2 W_B^2 + \cdots + s W_A^N W_B^N \tag{3}$$

where the sums are independent enough, meaning that $\text{rank}(W_A^i W_B^i) + \text{rank}(W_A^j W_B^j) \geq r$.

However, implementing restarts is not trivial in practice and requires certain modifications to the optimization procedure. Naïve implementation causes the model to diverge right after the restart. Unlike plain stochastic gradient descent, which solely relies on the value of the gradient at the current optimization timestep, Adam [29] update is guided mainly by the first and second moments of the gradient accumulated over the previous steps. In practice, gradient moment smoothing parameters $\beta_1$ and $\beta_2$ are usually very high $0.9 - 0.999$. Let's assume that at the reinitialization boundary $W_A^1$ and the corresponding gradient moments $m_A$ and $v_A$, are full-rank $(r)$. Then, after the merge-and-reinit, continuing to use old gradient moments for $W_A^2$ will guide it in the same direction as $W_A^1$ and optimize the same subspace.

To resolve this issue, we propose ReLoRA. ReLoRA performs a partial reset of the optimizer state during merge-and-reinit and sets the learning rate to 0 with a subsequent warmup. Specifically, we set 99% of low-magnitude optimizer state values to zero and use a jagged-cosine learning rate schedule (Figure 2). Our ablation studies (Section 3) show that both of these modifications are required to improve the performance over vanilla LoRA.

To reiterate, ReLoRA is a low-rank training method inspired by LoRA that uses restarts to increase the effective rank of the update, uses partial optimizer reset, and a jagged scheduler to stabilize training and warm starts. All of this allows ReLoRA to achieve performance comparable to full-rank training, especially in large transformer networks, by only training a small set of parameters at a time. ReLoRA is described in Algorithm 1.

**Enhancing computational efficiency**   Unlike other low-rank training techniques [44, 49], ReLoRA follows the LoRA approach by maintaining the frozen weights of the original network and adding new trainable parameters. At first glance, this may appear computationally inefficient; however, the differentiation between frozen and trainable parameters plays a crucial role in parameter-efficient fine-tuning [33].

These methods achieve significant improvements in training time and memory efficiency by reducing the size of the gradients and the optimizer states. Notably, Adam states consume twice as much memory as the model weights. Moreover, it is common practice to maintain gradient accumulation buffers in 32-bit precision for large networks, thereby adding significant overhead to the memory consumption of gradients.

By substantially reducing the number of trainable parameters, ReLoRA enables the utilization of larger batch sizes, maximizing hardware efficiency. Additionally, it reduces the bandwidth requirements in distributed setups, which are often the limiting factor in large-scale training.

Furthermore, since the frozen parameters are not being updated between restarts, they can be kept in a low-precision quantized format, further reducing their memory and computational impact. This additional optimization contributes to overall improved efficiency in terms of memory utilization and computational resources of ReLoRA and increases at scale.

# 4   Experiments

To evaluate the effectiveness of ReLoRA, we apply it to train a transformer language model on the C4 dataset [41] using various model sizes: 60M, 130M, 250M, and 350M. Language modeling has been shown to be a fundamental task in machine learning [40], it enables text and image classification [56], translation [8], programming [9], in-context learning, step-by-step reasoning [54], and many other emergent abilities [53]. Given its significance, we focus solely on language modeling for the purposes of this paper.

**Architecture and training hyperparameters**   Our architecture is based on transformer [51] and closely resembles LLaMA [50]. Namely, we use pre-normalization, RMSNorm [58], SwiGLU activations [45], $\frac{8}{3}h$ fully-connected hidden state size [50], and rotary embeddings [48]. All hyperparameters are presented in Table 1.

We use bfloat16 for all floating point operations and Flash attention [13] for effective attention computation. Compared to attention in LLaMA, which uses float32 for softmax computation, this increased training throughput by 50-100% without any training stability issues.

**Algorithm 1** ReLoRA. $\theta$ is model parameters, $\hat{\theta}$ is model parameters with linear layers replaced with ReLoRA, $M$ and $V$ are Adam optimizer states, $\eta$ is learning rate scheduled according to a jagged scheduler, and finally, $q$ is the reinit frequency.

---

**Require:** $\theta, M, V, q, \eta$
 1: **for** t **in** warm start steps **do**
 2:     Update $\theta$, $M$, $V$, $\eta$ {Regular training for warm start}
 3: **end for**
 4: **for** layer in model layers **do**
 5:     **if** layer **is** linear **then**
 6:         layer $\leftarrow$ ReLoRA($W^i, W_A^i, W_B^i$)
 7:         Freeze $W^i$
 8:     **end if**
 9: **end for**
10: **for** t in training steps **do**
11:     Update $\hat{\theta}$, $M$, $V$ {Training step with ReLoRA}
12:     **if** MOD$(t, q) = 0$ **then**
13:         **for** l in model layers **do**
14:             **if** l **is** linear **then**
15:                 $W^i \leftarrow (W^i + sW_A^i W_B^i)$
16:                 $W_A^i \leftarrow$ kaiming_init($W_A^i$); $W_B^i \leftarrow 0$
17:                 $M_{W_A^i} \leftarrow$ prune($M_{W_A^i}$); $V_{W_A^i} \leftarrow$ prune($V_{W_A^i}$)
18:             **end if**
19:         **end for**
20:         Start $\eta$ warmup
21:     **end if**
22: **end for**
23: **return** $\theta$

---

Most of our models were trained on 8 RTX 4090 for one day or less. Due to computational constraints, we train much smaller models than LLaMA, with the largest model having 350M parameters, the same as BERT Large [15]. We select the number of pre-training tokens based on the Chinchilla scaling laws [22] for all models, except for the largest one, which we train for 6.8B tokens while 9.5B tokens are Chinchilla-optimal.

**ReLoRA and baselines setup** In our low-rank training experiments, ReLoRA replaces all attention and fully-connected network parameters, while keeping the embeddings full-rank. The RMSNorm parametrization remains unchanged. Since ReLoRA-wrapped models have fewer trainable parameters than full-rank training, we include a Control baseline, which is a full-rank transformer with the same number of trainable parameters as ReLoRA.

We initialize ReLoRA from a checkpoint of full-rank training at 5,000 update steps and reset it every 5,000 steps thereafter, 3 times in total. After each reset, 99% of the optimizer state is pruned based on magnitude, and the loss is warmed up for the next 100 iterations. ReLoRA parameters are reinitialized following LoRA best practices, Kaiming initialization [20] for $A$-matrix, and zeros for $B$-matrix. In case of not using the restarts, the $B$-matrix also uses Kaiming initialization to avoid gradient-symmetry issues.

| Params | Hidden | Heads | Layers | Learning rate | Batch (tokens) | Seq. len. | Tokens |
|--------|--------|-------|--------|---------------|----------------|-----------|--------|
| 60M | 512 | 8 | 8 | 1e-3 | 122K | 256 | 1.2B |
| 130M | 768 | 12 | 12 | 1e-3 | 154K | 256 | 2.6B |
| 250M | 768 | 16 | 24 | 5e-4 | 590K | 512 | 6.8B |
| 350M | 1024 | 16 | 24 | 5e-4 | 590K | 512 | 6.8B |

Table 1: Hyperparameters of the language models trained in this study.

|                                       | 60M   | 130M  | 250M  | 350M  |
|---------------------------------------|-------|-------|-------|-------|
| Full training                         | 33.81 | 23.65 | 22.39 | 20.40 |
| Control                               | 36.52 | 27.30 | 29.12 | 23.65 |
| Low-rank pre-training with LoRA       | 47.44 | 34.17 | 36.60 | 57.11 |
| Low-rank pre-training with ReLoRA     | 38.28 | **25.04** | **23.28** | **22.48** |
| No. of training tokens (billions)     | 1.2   | 2.6   | 6.8   | 6.8   |

Table 2: Comparing perplexities between baseline methods and ReLoRA (lower is better). Control has the same number of trainable parameters as low-rank training. Low-rank training is **bold** if it outperforms the Control baseline. Notice that ReLoRA efficacy increases as the network size grows.

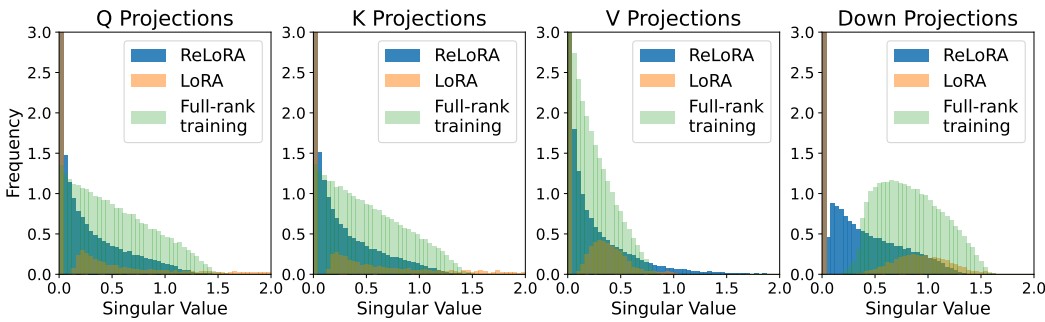

Figure 3: Singular values spectra of the weight difference between ReLoRA and LoRA at 5,000 iterations (warm start) and 20,000 iterations. ReLoRA exhibits a closer resemblance to full-rank training singular values than LoRA, indicating its effectiveness in approximating full-rank behavior.

## 5 Results

**Parameter-efficient pre-training** Our main results are resented in Table 2. ReLoRA significantly outperforms low-rank LoRA training demonstrating the effectiveness of our proposed modifications (ablated in Section 3). Furthermore, ReLoRA achieves similar performance to full-rank training, and the performance gap diminishes as network size increases.

Interestingly, the only model in which ReLoRA couldn't surpass the Control baseline was our smallest model with 60M parameters. This observation suggests that ReLoRA is particularly effective in improving the training of large networks, which aligns with our goal of developing a method that improves large-network training.

**High-rank training through low-rank updates** To determine whether ReLoRA performs a higher rank update than LoRA we plot the singular value spectrum of the difference between warm-start weights and the final weights for ReLoRA, LoRA, and full-rank training. Figure 3 illustrates significant qualitative differences between LoRA and ReLoRA for the singular values of $W_Q$, $W_K$, $W_V$, and $W_{down}$.

While most of the singular values for LoRA are zero (Figure 4) with a noticeable number of exceptionally high values above 1.5, ReLoRA exhibits a higher distribution mass between 0.1 and 1.0, reminiscent of full-rank training. This observation emphasizes the significance of high-rank updates and demonstrates the qualitative efficacy of ReLoRA, which accomplishes a high-rank update by performing multiple low-rank updates.

### 5.1 Ablation studies

We conduct ablation studies on all four crucial components of ReLoRA: restarts, jagged schedule, optimizer resets, and warm starts, utilizing the 130M-sized model. The results are presented in Table 3. In this section, we will focus on and analyze certain combinations of these components.

| Restarts | Jagged Schedule | Optimizer Reset | Warm Start | Perplexity ($\downarrow$) |
|:---:|:---:|:---:|:---:|:---:|
| $\times$ | $\times$ | $\times$ | $\times$ | 34.17 |
| $\checkmark$ | $\times$ | $\times$ | $\times$ | 34.25 |
| $\checkmark$ | $\times$ | $\checkmark$ | $\times$ | N/A |
| $\checkmark$ | $\checkmark$ | $\times$ | $\times$ | 34.29 |
| $\checkmark$ | $\checkmark$ | $\checkmark$ | $\times$ | 29.77 |
| $\times$ | $\times$ | $\times$ | $\checkmark$ | 25.46 |
| $\checkmark$ | $\checkmark$ | $\checkmark$ | $\checkmark$ | **25.04** |

Table 3: Ablation studies of ReLoRA. Restarts and warm starts are essential for good performance. Using restarts and optimizer reset without a jagged schedule causes the model to diverge.

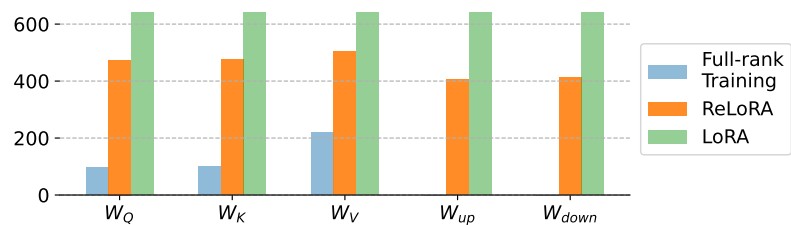

Figure 4: Number of singular values $< 0.1$ in attention and FCN projection matrices.

**LoRA**  ReLoRA, without the aforementioned components, is essentially equivalent to training a low-rank network parameterized by LoRA. This approach yields remarkably high perplexity, indicating that a simple matrix decomposition has significantly different training dynamics from full-rank training.

**Adding restarts and optimizer resets**  ReLoRA, without a jagged schedule and optimizer reset, performs similarly to LoRA because old optimizer states force the newly initialized parameters into the same subspace as the prior weights, limiting the model's capacity. However, doing a naive optimizer reset with ReLoRA causes the model to diverge. A jagged schedule helps to stabilize training and has a positive impact on the mixture. In our initial experiments, we also observed that a combination of partial optimizer reset and jagged scheduler allows for a quicker warm-up, as low as 50 steps, instead of hundreds of steps required when the optimizer is initialized from scratch.

**Warm start**  The warm start shows the most significant improvement, dropping perplexity by almost 10 points. To investigate whether post-warmup training contributes to the loss, we measured the perplexity of the warmed-up network, which equals 27.03. It outperforms all low-rank methods except for our final ReLoRA recipe but still demonstrates a significant difference from the final network. This demonstrates the importance of early training, similar to the concept of the lottery ticket hypothesis with rewinding [18].

## 6  Conclusion

In this paper, we investigated low-rank training techniques for large transformer language models. We first examined the limitations of a simple low-rank matrix factorization (LoRA) approach and observed that it struggles to effectively train high-performing transformer models. To address this issue, we proposed a novel method called ReLoRA, which leverages the rank of sum property to train a high-rank network through multiple low-rank updates. Similar to the lottery ticket hypothesis with rewinding, ReLoRA employs a full-rank training warm start before transitioning to ReLoRA. Additionally, ReLoRA introduces a merge-and-reinit (restart) strategy, a jagged learning rate scheduler, and partial optimizer resets, which collectively enhance the efficiency of ReLoRA and bring it closer to full-rank training, **particularly in large networks**. ReLoRA efficiency increases with the network size making it a viable candidate for multi-billion-scale training.

We firmly believe that the development of low-rank training methods holds great promise for improving the efficiency of training large language models and neural networks in general. Furthermore, low-rank training has the potential to provide valuable insights for the advancement of deep learning theories, aiding our understanding of neural network trainability through gradient descent and their exceptional generalization capabilities in the overparametrized regime.

## 7    Limitations and Future Work

**Scaling beyond 350M**    Due to limited computational resources, our experiments were constrained to training language models with up to 350M parameters. Nonetheless, ReLoRA already demonstrates promising results at this scale. However, we anticipate its true potential will be realized in the 1B+ parameter region. Additionally, while the 350M model outperforms the Control baseline, it does not continue the trend of narrowing the gap between ReLoRA and full-rank training. We attribute this to suboptimal hyperparameter choice, which requires further investigation.

Furthermore, in 60-350M experiments, even though ReLoRA significantly reduces the number of trainable parameters, we did not observe substantial improvements in memory and computation for the networks of this size. To evaluate the efficiency of our current implementation at a larger scale, we trained the 1.3B-parameter model for a small number of iterations to estimate memory and compute improvements of ReLoRA. At this scale, we observe *30% memory consumption reduction and 52% training throughput increase*. We expect to observe even bigger improvements over the full-training baseline for larger networks since the number of trainable parameters for ReLoRA, similar to LoRA, increases at a much slower rate compared to the number of frozen parameters. ReLoRA implementation could be further improved by effectively utilizing gradient checkpointing for ReLoRA layers, custom backward functions, and converting frozen model weights to int8 or int4 quantized format [14].

**Comparison to other low-rank training methods**    A number of approaches to low-rank training have been explored with other model architectures in earlier work [44, 49, 55]. Two aspects set our work apart from these earlier efforts. First, the approach we propose performs high-rank updates through low-rank training. Second, our work demonstrates competitiveness of the low-rank training methods in large-scale transformer language models with 100M+ parameters.

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
