# OpenReview forum: "Stack More Layers Differently: High-Rank Training Through Low-Rank Updates"
_NeurIPS.cc/2023/Conference — Submitted to NeurIPS 2023_

### Official Review · Reviewer_wgeh · 2023-06-28

**Soundness:** 2 fair
**Presentation:** 3 good
**Contribution:** 3 good
**Rating:** 4
**Confidence:** 5

**Summary:**

This paper proposed a low-rank way of training LLMs which allows more flexibility than most if not all popular low-rank training methods. It's a parameter-efficient way which only consumes limited gpu usage so it has the potential to be applied to larger models.

**Strengths:**

1. The rationale behind the idea is reasonable. We'd like to have a higher rank models to provide capacity and in the meantime keeping # trainable parameters small.

2. The Introduction reads well.



**Weaknesses:**

Overall, I like the idea but somehow there are few concerns that I think needs to be improved in order to accept it.

1. It's not well written as many there are many notation unexplained (L85 r), linkage error (e.g., L106 sec 3), and the algorithm is not well explained in a line by line fashion. Many details are skipped, which leads to the problem that reading the code is the only way to fully capture the proposed method.

2. The motivation is not convincing. Rank(A + B) < Rank (A) + Rank(B) doesn't guarantee Ranks(A + B) will be larger than min( Rank(A) + Rank(B)). Without any further constraints, there is no guarantee L93 is true. We just know the upper bound is higher but to claim it, we need to bound lower-bound, which is not discussed at all. The idea itself is interesting but rationale is wrong.

3. Experiments are limited, as shown in Question sections.

**Questions:**

1. A fundamental problem with the setup is that LoRA is used for fine-tune a pre-trained LLM "for downstream tasks". Thus, comparing PEFT methods on pre-training performance seems to be meaningless. Authors should compare ReLORA for both pre-train + fine-tuning vs full train + LoRA fine-tune and analyze the ranks or so to make readers understand what's going under the hood. Without this type of analysis (no matter ReLORA performs better or worse), the insights are limited.

2. Following 1, I'd like to see some analysis/experiments on downstream tasks like GLUE/super-GLUE tasks.

3. Jagged Schedule is rather arbitrary. Not sure what's going on if I switch to another dataset, switch models and how it should be combined with fine-tuning. There is no theoretical analysis of the stability, nor experimental experiments to demonstrate it.

4. Another baseline which should be included is training using plain SGD with Jagged Schedule. The above mentioned problems with Adam might totally be redundant given such a complex schedule designed.

5. What's the formal definition of Control? It's not clear how it's performed and frankly speaking, the performance of control is fairly comparable to ReLoRAin Table 2.

6. Selection of r needs to be discussed.

**Limitations:**

It's discussed in the paper and it reads well.

---

> ### Author Rebuttal · Authors · 2023-08-09
>
> Thank you for your detailed feedback and thorough review of our paper. We sincerely appreciate your insights and would like to address your concerns. Here's our response to your comments:
>
> Regarding your statement about PEFT methods: the goal of this paper is to demonstrate that parameter-efficient methods can be effectively used at the resource-demanding and expensive pre-training stage. Comparing ReLoRA for both pre-train + fine-tuning vs. full train + LoRA is definitely interesting, but our goal was to develop a way to use parameter-efficient methods to increase the efficiency of pre-training.
>
> ### Motivation Concerning Ranks
>
> While there is no guarantee that Rank(A+B) is greater than the minimum of the two ranks (in fact, without additional constraints, there is no guarantee that it is greater than zero), our goal in this section was to provide an intuition for why ReLoRa restarts can increase the rank of the update. We do not claim a theoretical result that ReLoRA yields a higher-rank update. However, we empirically demonstrate that ReLoRA increases the rank of the learned update compared to LoRA (see Figures 3 and 4 in the paper). Our intuition concerning the Adam state affecting the performance of ReLoRA is verified in our ablation study.
>
> ### Jagged Schedule
>
> We base our schedule on a well-established linear warmup + cosine decay schedule used, for example, in BLOOM, LLaMA, and Pythia models [BigScience Workshop, 2022; Touvron et al., 2023; Biderman et al., 2023]. The "jaggedness" we introduce is not arbitrary and is aimed to avoid loss divergence, as we demonstrate in our ablation study.
>
> ### On the Use of SGD
>
> Adaptive learning rate optimizers such as Adam are essential to training large neural networks because they mitigate saddle points [Staib et al., 2019], which cannot be mitigated with a learning rate schedule. During ReLoRA development, we considered using second-order deep learning optimization methods such as Shampoo [Gupta et al., 2018]. However, just like Adam, these optimizers are stateful; thus, they exhibit the same issues as Adam in the case of ReLoRA.
>
> ### Selection of r
>
> The value "r=128" was determined based on our preliminary experiments. Similar to LoRA, we observe little difference between the performance of methods trained with reasonable r. The main distinction is that in fine-tuning on a small dataset, r=16 or even r=8 could be used, but values below 64 demonstrated poor performance. Another consideration for the selection of r – it shouldn’t be more than half the hidden size as, at this point, the number of trainable parameters of full-rank and LoRA-based networks matches.
>
> ### Addressing presentation issues
>
> The term r at L85 is defined in equation 2 between the lines 85 and 86. We will make this definition more explicit. We apologize for the typo at L106; the reference was intended for "Table 3" instead of "Section 3". We deeply care about the paper's clarity, so please let us know if there are other notations or sections that need further clarification.
>
> ### Other comments
> We appended our fine-tuning results to the rebuttal for further clarity.
>
> The "Control" baseline is essentially a transformer network trained conventionally (not in low-rank) but with the same parameter count as the number of trainable parameters of the corresponding low-rank method.
>
> Thank you for your valuable feedback. We believe it will significantly improve our paper. We would be grateful for a reconsideration of our paper's score, as we feel our work introduces a novel and impactful result that reparametrization-based PEFT methods can be used to reduce the costs of an expensive pre-training stage.

---

> > ### Comment · Reviewer_wgeh · 2023-08-20
> >
> > Thanks for your reply. But I think my major concerns are not alleviated.
> >
> > 1. Comparing ReLoRA for both pre-train + fine-tuning vs. full train + LoRA is "needed" IMO as we want to know what's the losing expressibility in this case. Your claim is not very convincing to me.
> >
> > 2. GLUE results are skeptical to me. There is no description on how the stuff is run and what's the setup for each category. In particular, the numbers didn't quite match the well-known results. Take MRPC for example, you can easily find out some numbers about .86ish (https://huggingface.co/Intel/bert-base-uncased-mrpc#:~:text=It%20achieves%20the%20following%20results,Accuracy%3A%200.8603) but authors reported .8038. I have no idea how experiments are done.
> >
> > 3. I think the responses on other technical details are reasonable. However, I'd like to see how these integrated into the main text and see how that reads to decide if the score could be increased.
> >
> >
> > Overall I agree with what authors claimed about the novelty and I did indicate that . But somehow that's the only merit and the whole paper is far from ready in terms of readability and soundness of experiments. It seems like other reviewer (Reviewer 4sGe) also feel the experiments are not well established. I encourage the authors to further revise the writeup for a better and easier reading experience which could really benefit the community.

---

> > > ### Author Response · Authors · 2023-08-21
> > >
> > > Thank you for your reply. We really appreciate having your perspective, but we don’t believe the comparisons you suggested are relevant for the study. We explain the details below.
> > >
> > > ### ReLoRA (pre-train + fine-tune) vs. full train + LoRA fine-tune:
> > > > 1. Comparing ReLoRA for both pre-train + fine-tuning vs. full train + LoRA is "needed" IMO as we want to know what's the losing expressibility in this case. Your claim is not very convincing to me.
> > >
> > > It’s not clear what additional hypothesis would be tested by comparing [ReLoRA pre-train + ReLoRA fine-tune] and [full-rank pretrain + LoRA fine-tune] – and whether this hypothesis would be falsifiable.
> > >
> > > Table 4 in our response already has an apples-to-apples comparison with full-rank training: you can see that [ReLoRA pre-train + full-rank fine-tune] and [full-rank pretrain + full-rank fine-tune] are competitive in all tasks we tested. We don’t see how adding the experiment you suggested can provide additional insight.
> > >
> > > We appreciate your concern with better understanding the implications of full-rank vs. low-rank training. This is why we looked at the qualitative and quantitative differences in the singular value spectra in Figures 3 and 4. This analysis clearly shows that the ReLoRA spectrum is more similar to full-rank training than to LoRA.
> > >
> > > ### GLUE Results Discrepancy
> > > > 2. GLUE results are skeptical to me. There is no description on how the stuff is run and what's the setup for each category. In particular, the numbers didn't quite match the well-known results. Take MRPC for example, you can easily find out some numbers about .86ish but authors reported .8038. I have no idea how experiments are done.
> > >
> > > BERT-base model was pre-trained on 128B tokens (accounting for epochs), while our models were trained on 7B tokens. Thus, the difference in the absolute performance of our models is expected and can be unequivocally attributed to more than **10X difference in compute**. Apples-to-apples comparison between models in our study clearly demonstrates that ReLoRA-pretrained models are competitive with full-rank pre-trained models trained on the same amount of data.
> > >
> > > We used the standard settings for GLUE in all experiments: 3 epochs, lr=2e-5 linearly decaying to zero, no weight decay, batch size 64. We will include all the details into the camera-ready.
> > >
> > > Please keep in mind that our goal in this study was not to establish SOTA, but to demonstrate the viability of a new pre-training method. We will further clarify this point in the camera-ready.
> > >
> > > ### Incorporation of Review Feedback
> > > > I think the responses on other technical details are reasonable. However, I'd like to see how these integrated into the main text and see how that reads to decide if the score could be increased.
> > >
> > > We would love to share the updated version with you, in order to evaluate the changes, but the NeurIPS policy **does not allow for it**. Unlike journals, the standard practice in top ML conferences is to rely on the authors' commitment to integrate the updates during the camera-ready phase.
> > >
> > > > the whole paper is far from ready in terms of readability
> > >
> > > We have put a lot of work into ensuring the clarity of the paper, and we would gladly incorporate any additional specific feedback.
> > >
> > > > It seems like other reviewer (Reviewer 4sGe) also feel the experiments are not well established
> > >
> > > As we note in our response to Reviewer 4sGe, we believe there was some confusion regarding our methods. Reviewers that responded during the rebuttal period seem to be satisfied with the additional results we provided.
> > >
> > > We value your feedback and will address your remaining concerns in our final submission. We hope this allows you to reconsider your score.

---

### Official Review · Reviewer_4sGe · 2023-07-05

**Soundness:** 2 fair
**Presentation:** 1 poor
**Contribution:** 2 fair
**Rating:** 3
**Confidence:** 3

**Summary:**

This paper focuses on low-rank training techniques and introduce ReLoRA that uses low-rank updates to train a high-rank network (<=350M parameters). The main idea is to employ LoRA during training and "restart" it in order to artificially increase the rank, which is a nice idea. The difficulty remains in the optimization process due to the gradient after the "reset". ReLoRA performs a partial reset of the optimizer state during merge-and-reinit and sets the learning rate to 0 with a subsequent warmup.

The proposed method is elegant and novel. It is a nice trick to increase artificially the rank of LoRA.

Overall, it is unclear to me what we mean by "pre-training". From the text, it seems we are talking about fine-tuning large models with adapters. In the experiment section, the authors talk about training on C4 with a model similar to LLaMA, which indicates pre-training. However, ReLoRA is initialized from a full-rank training at 5k steps. My guess would be that the authors initialize randomly a model (e.g., BERT) and pre-train using an adapter approach.

I am disappointed with the experiment section. While I understand the limitation in resource computation, training models <= 350MB is disappointing when playing in the league of training large neural networks (as suggested by the title) or comparing with LLaMA. Nowadays, 8  GPU days of compute allows to fine-tune a LLaMA or similar model (in 1 day), not to pre-train.
In Table 2, it is missing the training time for each model to understand the benefits of ReLoRA over the control, full training, and LoRA. If the method is 2 times slower than Control, it does not necessarily makes sense to use it since the performance at 350M is less than 1ppl (which I expect to be even smaller with larger models).

Evaluating LMs only on perplexity is not sufficient. Please compare pre-trained models on GLUE/etc (as commonly done for adapters/prompting papers) using fine-tuning and PEFT approaches. This will show that:
- The base model after pre-training with ReLoRA is (maybe?) better than full-rank pre-training or standard pre-training
- Higher performance can be obtained with ReLoRA during fine-tuning.

missing references:
[1] Wang et al. 2023, LEARNING TO GROW PRETRAINED MODELS FOR EFFICIENT TRANSFORMER TRAINING (ICLR)

**Strengths:**

- Nice trick to increase artificially the rank of LoRA
- The proposed method is sound

**Weaknesses:**

- the paper writing and clarity must be improved
- the experiment section is insufficient


**Questions:**

See above

**Limitations:**

Yes

---

> ### Author Rebuttal · Authors · 2023-08-09
>
> Thank you for your feedback. Your comments have helped clarify certain areas of the paper that we need to address. Here's our response to your points:
>
> ### Clarification on Pre-training vs. Fine-tuning:
> It seems there is some confusion regarding the distinction between pre-training and fine-tuning, which is important here.  Pre-training involves training with a language modeling objective on large amounts of free text, rather than fine-tuning on a relatively small specific downstream task data. To the best of our knowledge, no prior work has applied parameter-efficient training methods to pre-training.
>
> We use the LLaMA architecture (a specific variation of transformer decoder), but our models are trained from scratch using the ReLoRA method. The ReLoRA training method includes a notably short warm start period, which is especially evident in our training cost estimates for a 1B model (see our reply to Reviewer 2, nJNf). Our models are trained from scratch using ReLoRa training, and there's no involvement of adapter techniques as Adapters [Houlsby et al., 2018] and LoRA [Hu et al., 2020] are conceptually different kinds of parameter-efficient fine-tuning methods (please refer to the survey by Lialin et al., [2023] for details). We appreciate your observation and will ensure this distinction is clear in the revised paper.
>
> ### Experiment Section & Model Size
> While we demonstrated the efficacy of our method at the scale of 350M parameters, we are optimistic about its scalability to larger networks as we found ReLoRA to be **more efficient in 250M and 350M** than in 70M and 130M networks (judged by the difference between full-rank and low-rank training). We recognize that our resource constraints limit us in some regards. However, we believe the review process should primarily focus on the soundness of the research rather than computational resources, and appreciate your understanding.
>
> ### Evaluation Metrics beyond Perplexity
> Although our primary focus was on showcasing computational efficiency during pre-training on a massive dataset, we agree that a comprehensive evaluation should encompass downstream tasks. To this end, we performed a **downstream evaluation on GLUE** tasks which you can find in the one-page PDF attached to the rebuttal. This additional evaluation reiterates that our method's performance is on par with traditional training, especially when considering computational savings.
>
> We appreciate the **reference suggestion** and will incorporate it into our related work section.
>
> We value your feedback and believe it will significantly improve our paper. Your observations have enabled us to present our findings more clearly, and we hope the revisions will address your concerns. We would be grateful for a reconsideration of our paper's score, as we feel our work introduces a valuable perspective to the deep learning community.

---

### Official Review · Reviewer_nJNf · 2023-07-06

**Soundness:** 3 good
**Presentation:** 3 good
**Contribution:** 2 fair
**Rating:** 5
**Confidence:** 3

**Summary:**

This paper introduces a novel approach, ReLoRA, for training large-scale neural networks. Recognizing the limitations of conventional low-rank matrix factorization (LoRA) in training high-performing transformer models, the authors propose ReLoRA that employs a high-rank network training through multiple low-rank updates. This new method uses a full-rank training warm start followed by a merge-and-reinit (restart) strategy, jagged learning rate scheduler, and partial optimizer resets, making it efficient particularly for large networks. The research finds that the efficiency of ReLoRA increases with the network size, positioning it as a potential candidate for efficient training of multi-billion-parameter networks. The paper's results suggest that low-rank training methods can potentially improve the efficiency of training large language models and provide valuable insights for deep learning theories. These insights could further our understanding of neural network trainability and their exceptional generalization capabilities in the overparameterized regime.

**Strengths:**

1. Important work of applying LoRA in pre-train
2. Good reproducibility: Code is released for readers; Hyperparameter settings are available. But still some parameters are missing, see below.
3. Sophisticated design of ablation study

**Weaknesses:**

1. What's the number of trainable parameters in the experiments? Rank r of LoRA is also not reported (or hard to find). I assume 60M, 130M, 250M, and 350M are the total number of parameters.
2. The perplexity reported for Control with 250M parameters appears to be an outlier, greater than that of 130M parameters. Please double-check. If it’s real, then the fluctuation range of perplexity could be too large for us to draw reliable conclusions with the results.
3. One important missing baseline is LoRA with warm-start, which is reported in the ablation study, but not in the main results of Table 2, as well as in Figure 3 and 4. As the gain of the main techs developed in the work for ReLoRA can only be seen when compared to the proper baseline of LoRA, unfair to compare with LoRA with no warm-start.

**Questions:**

1. Can authors also report the absolute amount of training resource saved (e.g. total GPU memory, total training wall time for each setting, instead of just 30% memory reduction and 52% training throughput)? Hard to see efficiency of low rank training without these values. Also curious to see how much more cost compared to LoRA with warm start, if there is any.

**Limitations:**

1. Initial warm start is important/indispensable to reach a performance comparable to the full training. If one start from scratch, with no warm started checkpoint, they may still be limited by the computing resource to warm start. The proposed method may just save some time later after warm start, which is, however, not shown in the results.

---

> ### Author Rebuttal · Authors · 2023-08-09
>
> Thank you for your assessment of our work. We really appreciate your feedback and suggestions! Here's a response to your questions:
>
> ### Trainable Parameters and Rank 'r' of LoRA
>
> You're correct about the parameter counts. We provide the total number of parameters (60M, 130M, etc.) in the paper. To specify, the **trainable** parameters for each low-rank model are:
> * 60M: 42M
> * 130M: 72M
> * 250M: 99M
> * 350M: 125M
>
> We updated Table 2 to make this clear (see attached PDF). Regarding the rank of LoRA, we now state in the **"Architecture and training hyperparameters"** section that for all LoRA and ReLoRA experiments, we use a rank of r = 128 based on our preliminary experiments.
>
> ### Perplexity Discrepancy for Control with 250M Parameters
> We checked the results, and there was an error in the hyperparameters, specifically, the number of layers in the model and lr warmup. The updated value for the Control with 250M parameters is 25.43, which is now in-between 72M and 99M models (control for 130M and 250M), as expected.
>
> ### Warm-Start LoRA Baseline:
> We added the warm-start + LoRA baseline to Table 2 for each model size and observed small but consistent improvement of ReLoRA over this baseline. We also performed additional experiments with a smaller warm-start phase to further validate our findings. Specifically, when restricting the warm-start phase to just 2K steps for the 350M model, we observed a gap of more than 1.4 ppl point between LoRA (25.08) and ReLoRA (23.64). While the absolute performance of ReLoRA is lower compared to full-rank training in this context, this experiment shows that LoRA restarts positively impact model performance.
>
> For all of these results, see the attached PDF.
>
> ### Absolute Training Resources:
> To provide a clearer perspective on the efficiency of our method, we estimated the costs for training a 1B model on 20B tokens (~Chinchilla-optimal) using a 2x3090 setup. Both models use sequence length 512 and microbatch size of 4 examples. The total batch size is 1152 examples or ~600K tokens.
>
> **Regular Training:**
> * GPU Memory: 21Gb per GPU
> * Throughput: Estimated at approximately 4,500 tokens/second
> * Wall time: **1235 hours** on 2x3090 GPUs
>
> **ReLoRA:**
> * GPU Memory: 14.5Gb per GPU
> * Low-rank training throughput: ~9,200 tokens/second
> * Wall time: **762 hours** on 2x3090 GPUs
> * Breakdown: Warm start phase (based on 25% of total training steps) takes around 309 based on 4.5K tokens/second full-rank training throughput estimate above. The subsequent low-rank training phase accounts for the remaining 453 hours with an estimated throughput of 9.2 K tokens/second.
>
> A noteworthy aspect of ReLoRA is the reduced GPU memory requirement. This allows for an increased microbatch size during training, contributing to enhanced efficiency and throughput during the low-rank training phase.
>
> ### Cost Comparison with Warm-Start LoRA:
> The difference in costs between LoRA and ReLoRA is minimal. The main overhead in ReLoRA is resetting the optimizer states, which doesn't add significant time to the training (only a few seconds).
>
> We hope these updates address your concerns. Your feedback has been crucial in refining the paper, and we appreciate it. We hope the changes made address your concerns, and we humbly request a reconsideration of the paper's score. As this work presents the first proposal for using parameter-efficient methods for pre-training, we believe it opens a promising avenue for new research, and we would greatly value the opportunity to share it with the NeurIPS community.

---

> > ### Comment · Reviewer_nJNf · 2023-08-16
> >
> > Thank authors for replying to all my questions and concerns. The results are now more reasonable and solid than in the initial version. Please integrate them in the final version. I'll also raise my rating.

---

### Official Review · Reviewer_Cooz · 2023-07-28

**Soundness:** 3 good
**Presentation:** 3 good
**Contribution:** 3 good
**Rating:** 5
**Confidence:** 4

**Summary:**

The paper proposes an extension LoRA. The main insight in the paper is  LoRa ca be initialization multiple times during training layers and this in the end will produce a high rank update. The authors show that it is quite challenging to re-intialize the layers mostly due to the internal initialization state of Adam.

The authors then propose a pruning based technique to overcome this limitation when using Adam. The also propose a learning rate schedule which helps to avoid the divergence of the network. The authors show that this leads to better performance than Lora on the C4 datasets.

**Strengths:**

The paper proposes an interesting technique which might be useful.

The experiments performed by the authors are reasonable and even with limited compute resources they clearly show a clear picture.

The authors have used the C4 dataset. Which is quite reasonable for a start

**Weaknesses:**

* The methods proposed is more of set of approaches to avoid divergence due to Adam.

* The techniques used a bit of trial and error, is there more principal.

* Would have been better to see what is the downstream performance of ReLoRa transform.

* Can you provide a better understanding why the second to last row in Table 3 performs very similar to No attemto

**Questions:**

Please look at the weakness section.

I am happy to bump up my score to a weak accept if you can do the following two things -

1. Perform evaluation on downstream task atleast one or two datasets

2. Why is performance so close in Table 3.



**Limitations:**

I think the paper is well written. I understand the lack of compute to perform large experiments.

---

> ### Author Rebuttal · Authors · 2023-08-09
>
> We appreciate your thorough review and the feedback provided. Thank you!
>
> To address your questions, we’ve performed the following additional experiments:
>
> ### 1. Performance Similarity in Table 3:
>
> You pointed out the similarity in performance between the second to last row in Table 3 and our baseline. To dig deeper into that, we performed two additional sets of experiments:
>
> **1.1. Warmup + LoRA baselines for all runs in Table 2:** After hparam tuning of both LoRA and ReLoRA, we confirm a small but consistent improvement of ReLoRA over warmstart+LoRA.
>
> **1.2. Smaller warm start experiments:** We conducted experiments with both LoRA and ReLoRA for the 350M model, but restricted the warm-start phase to 2K steps. The results show a performance gain with ReLoRA over LoRA by 1.4 ppl points (ppl 23.64 vs 25.08). While the absolute performance of ReLoRA is lower compared to full-rank training in this context, these experiments validate our initial hypothesis that LoRA restarts positively impact performance.
>
> These experiments show that ReLoRA offers consistent improvements over warmed-up LoRA, shedding light on the distinction between the two methodologies.
>
> ### 2. Downstream Performance Evaluation:
>
> Based on your recommendation, we attached supervised fine-tuning results for full-rank, ReLoRA, and non-pretrained 350M models. ReLoRA shows downstream performance on par with full training, beating the model finetuned from random initialization baseline.
>
> Please note that absolute performance is lower than e.g. BERT, since BERT is trained 128B tokens compared to 7B in our experiments.
>
> For all of these results, see the single-page PDF attached.
>
> We believe our results demonstrate the potential of ReLoRA as a next generation method for model training, and specifically, that parameter-efficient methods can be applied at the resource-heavy pre-training stage. We hope that the additional experiments and clarifications address your concerns and make a compelling case for this paper. Thank you once again for your insights, and we look forward to the committee's feedback.

---

### Author Rebuttal · Authors · 2023-08-09

We sincerely appreciate the time and effort the reviewers dedicated to reviewing our paper. Your feedback, ranging from detailed concerns to constructive suggestions, has been instrumental in guiding us to refine and clarify our work.

Following the NeurIPS rebuttal policy, we attach a single-page PDF with additional experiments. It includes one figure and two tables.

* Table 2 is updated with warm start + LoRA baseline for every model and also provides the number of trainable parameters.
* Figure 5 demonstrates a significant difference between LoRA and ReLoRA warm started only after 2K steps.
* Table 4 provides a downstream evaluation on several GLUE tasks.

Given the clarifications provided in answers to reviewers individually and the additional experiments presented in the attached one-page PDF, we kindly request a re-evaluation of our manuscript's scores.

Thank you for the time and effort dedicated to reviewing our work.

---

### Decision · Program_Chairs · 2023-09-21

**Decision:**

Reject

**Comment:**

This paper proposes an approach (ReLoRa) that uses low-rank updates to efficient train large neural networks. The reviews for the paper were mostly borderline to leaning negative with primary concerns being: (1) weak empirical analysis (which the authors partially address) (2) somewhat heuristic nature of the approach and lack of theoretical basis (3) lack of proper ablation (both of which I think are valid points). I think the approach is interesting but I believe that the paper will need significant revision to address these issues. For these reasons, I recommend rejection but I strongly encourage the authors to submit a revised version to a future conference.